# Determination of a New Coastal ENSO Oceanic Index for Northern Peru

Edgard Gonzales [1,2,*] and Eusebio Ingol [1]

1   Universidad Nacional Agraria La Molina, La Molina, Lima 15026, Peru; eingol@lamolina.edu.pe
2   Universidad Nacional de San Agustín de Arequipa, Calle Santa Catalina 117, Cercado, Arequipa 04000, Peru
*   Correspondence: hgonzalesz@unsa.edu.pe; Tel.: +51-54-959341704

**Abstract:** In 2017, extreme rainfall events occurred in the northern portion of Peru, causing nearly 100,000 victims, according to the National Emergency Operations Center (COEN). This climatic event was attributed to the occurrence of the El Niño Southern Oscillation (ENSO). Therefore, the main objective of this study was to determine and differentiate between the occurrence of canonical ENSO, with a new type of ENSO called "El Niño Costero" (*Coastal El Niño*). The polynomial equation method was used to analyze the data from the different types of existing ocean indices to determine the occurrence of ENSO. It was observed that the anomalies of sea surface temperature (SST) 2.5 °C (January 2016) generated the "Modoki El Niño" and that the anomaly of SST −0.3 °C (January 2017) generated the "Modoki La Niña"; this sequential generation generated El Niño Costero. This new knowledge about the sui generis origin of El Niño Costero, based on the observations of this analysis, will allow us to identify and obtain important information regarding the occurrence of this event. A new oceanic index called the Pacific Regional Equatorial Index (PREI) was proposed to follow the periodic evolution and forecast with greater precision a new catastrophic event related to the occurrence of El Niño Costero and to implement prevention programs.

**Keywords:** ENSO; Modoki El Niño; Modoki La Niña; El Niño Costero; Pacific Regional Equatorial Index; Peru

## 1. Introduction

Globally, El Niño Southern Oscillation (ENSO) events are one of the most important sources of annual climate variability. These phenomena involve large-scale interactions between the oceans and the atmosphere, which consist of an oscillation of atmospheric pressure in the western Pacific [1]. The relationship or coupling between these variables involves great climatic consequences in many parts of the planet [2]. The occurrence of ENSO in the tropical Pacific causes a variation on the oceanic-atmospheric systems in both the North Pacific and the North Atlantic [3]. Without going any further, ENSO is considered the most powerful signal in terms of the interannual variation of the oceanic-atmospheric system [4].

Records indicate that the occurrence of ENSO in South America has shown statistically significant evidence since 1750. However, from the 18th century onwards the chronology of ENSO and anti-ENSO events have occurred with similar chronologies in the literature [5]. Evidence of the widespread impact of ENSO-related events is associated with variables such as (1) sea surface temperatures (exceeding 28 °C); (2) increased rainfall amounts in the central Pacific region; and (3) storm surge in subtropical North and South America. ENSO events, on the other hand, are also associated with droughts in many parts of the world [6,7].

Recent studies focusing on ENSO have shown that changes in the thermocline's depth (i.e., transition layer between warmer water at ocean surface and cooler water deeper down) are the main cause of sea surface temperature (SST) variations, in the time scale and, consequently, the generation of ENSO events [8]. Models made between ocean and

atmosphere can reproduce the aperiodic oscillation of the Pacific between warm states (El Niño) and cold states (La Niña), with an average return interval of 3–4 years [9,10]. El Niño and La Niña are opposite phases of what is known as the "ENSO cycle".

### 1.1. The Effects of the El Niño Southern Oscillation (ENSO) in Peru

In Peru, ENSO manifests as rainfall increases in the coastal area of the country, which causes flooding, destruction of infrastructure, and deaths, in addition to generating significant contractionary effects on the fishing, agricultural, and primary manufacturing sectors. The occurrence of ENSO in this South American country has had significant effects during the periods 1982–1983, 1986–1987, 1991–1992, 1997–1998, and 2016–2017 [11]. Specifically, the ENSO phenomenon corresponding to the period 1982–1983 was significantly intense, producing catastrophic losses [12]. In fact, in northern Peru the climatic alteration caused by this phenomenon manifested itself with severe droughts in the south and in the Altiplano Region of the country, severely affecting the nation's socioeconomic activities, being classified as an "extraordinary event" [12]. The 1997–1998 ENSO, on the other hand, generated rainfall in the extreme north of the country (Apurímac, Ayacucho, La Libertad, Lambayeque, Piura, and Tumbes Departments), showing very high levels compared to normal, and more intense than those that occurred during the 1982–1983 ENSO [13].

Intense ENSO-related rainfalls occurred in northern Peru during January, February, and March of 2017, exceeding records documented during the 1982–1983 ENSO, affecting cities such as Tumbes, La Libertad, Ancash, Lambayeque, and Piura [14]. The anomalous warming generated on the coasts of northern Peru during those months was different from the typical ENSO-developing conditions, although its manifestation was similar [11].

Many studies have focused on the occurrence of ENSO in Peru; reference [15] for example, evaluated the behavior of the ENSO phenomenon in the eastern and central Pacific, concluding that it has different impacts on South America's rainfall (including Peru) and on the atmospheric trajectory through the South Pacific Convergence Zone (SPCZ) and the Inter-Tropical Convergence Zone (ITCZ), both of which are poorly understood. The authors also reanalyzed data during the austral summer (December through February) for the 1980–2016 period, determining dry anomalies along the tropical Andes and northern South America, while humid anomalies prevailed in south-eastern portion of the continent; reference [16] on the other hand, indicated that the strong and warm 1997–1998 ENSO that occurred at the Independencia bay (Peru), showed an increase in sea surface temperature of 10 °C, presenting higher concentrations of oxygen and clearer water, due to the decrease in the phytoplankton concentration.

Despite the aforementioned studies, current knowledge about the behavior of the ENSO phenomenon and its associated meteorological anomalies before the beginning of instrumental records is deficient [17]. Therefore, it is unknown whether it is a significantly stable phenomenon, or if there have been events in the past where it was more inactive or active than it is today. Similarly, it is unknown whether ENSO can permanently shift into an El Niño or La Niña cycle. Investigating these questions has implications both in the reconstruction of past climates and in the prediction of future climatic variations. In addition to the need for understanding of how ENSO works, there has been a variation of ENSO, called "El Niño Costero" (2017 Coastal ENSO), whose occurrence is not common, and its generation process is not well studied.

### 1.2. Development of the Coastal ENSO-2017 in Peru

In the 2017 event, a teleconnection between the western Pacific and the extreme east was evidenced as a trigger mechanism. Using the National Centers for Environmental Predictions NCEP2 reanalysis, a deep convection was found over central Australia in January 2017 triggering Rossby wave trains that spread across the South Pacific modifying the tropospheric west flow and impacting the subtropical Andes [18]. This teleconnection showed improved deep convection (i.e., more precipitation) over the eastern Amazon and

reduced deep convection (i.e., less precipitation) in the western tropical Pacific during spring summer reducing wind surface [18].

Coexisting SST anomalies in the equatorial Pacific (and presumably also in the Tropical Atlantic) clearly favored the development of the extreme "coastal El Niño" event and high-magnitude rainfall over Peru in DJFM 2016–2017. The approach and interpretation within a climatic context, shows that the DJFM 2016–2017 rainfall pattern in Peru was highly anomalous, both in terms of its magnitude and the time after a strong El Niño event, which in turn was largely the result of abrupt and unexpected warming above average in the Niño 1 + 2 region. This warming was linked to an anomalous weakening of the western mid-upper level subtropical flow, which in turn led to a weakening of the southeast winds off the coast favoring the warming of the eastern Pacific [19]. This flow is a key ingredient in the subtropical circulation of the SE Pacific, because its blockage by the subtropical Andes leads to a decrease in the flow towards the Equator [18].

A reduction in deep convection in the western central Pacific and zonal air humidity flow in the tropical troposphere may have shifted the ITCZ southward, increasing precipitation along the Andes of northern Peru [15,20]. The Eastern Tropical Pacific also experienced rapid and marked warming in early 2017, causing torrential rains along the west coast of South America with significant social cost in Peru and Ecuador. This strong coastal El Niño was largely unpredictable even a few weeks before its onset, and developed differently than the central or eastern events with a sustained and forced weakening of the western tropospheric free flow impinging on the subtropical Andes driving to a relaxation of the southeast (SE) wind off the coast, which in turn may have warmed the eastern Pacific [20].

A transformation of the El Niño teleconnections has occurred due to climate change and the effect it may have had in the extremely dry conditions during 2015/16 than during 1982/83 and 1997/98. Such a transformation could arise from changes in the atmospheric circulation of forced tropical wave trajectories [21], or from changes in the intensity and length of rainfall from the equatorial Pacific during El Niño events [22].

The occurrence of a coastal ENSO is not common; for example, previous similar event occurred in 1925, that is, it occurred after 92 years and it is stated that its internal dynamics coupled with the ITCZ region was largely the cause for the coastal El Niño of 1925 in the absence of significant external atmospheric forcing [20].

Ocean and atmospheric modeling experiments show that the combined effect of local winds and equatorial Kelvin waves caused the extreme coastal El Niño of 2017, amplified by a positive feedback from Bjerknes that is very important to the extreme coastal ENSO of 2017. Additionally, pre-existing SST warming along the west coast of subtropical South America also favored anomalous north winds. These wind anomalies were favorable for the warming of the coasts south of the equator through weakened upwelling and wind-evaporation-SST (WES) feedback, setting the stage for the dramatic growth of the extreme coastal ENSO. Furthermore, extreme coastal ENSO frequency changes are independent of future basin-scale changes [23].

After the occurrence of ENSO 2015/2016, the surface of the eastern Pacific Ocean remained ∼0.5 °C warmer than average, experiencing an additional 1.5 °C warming in a series of one-week episodes of duration in the months of January, February and March 2017, which caused torrential rains along the west coast of South America. The behavior of the ocean-atmosphere system during this event differs substantially from the canonical dynamics of ENSO, and a dominant condition of external atmospheric forcing prevails, as well as a sustained and forced weakening of the western tropospheric flow that affects the subtropical Andes, leading to a relaxation of southeast (SE) winds off the coast, which in turn may have warmed the eastern Pacific along weakening outcrops in a near-shore band and decreasing evaporative cooling further offshore [24].

For this reason, this investigation seeks to understand the coastal ENSO and its effects, since there is currently no precise information on the cause that generates it, as well as the interaction with other similar meteorological phenomena that enhance its effects. Knowledge of the occurrence of coastal ENSO would be of great importance for

the prevention, planning, and management of agriculture, water resources, and natural disasters. Thus, the objective of this study is to determine the occurrence of ENSO, but as a variation defined as "2017 Coastal ENSO". Therefore, our focus was mainly to analyze the different types of oceanic indices to determine the occurrence of coastal ENSO.

## 2. Materials and Methods

### 2.1. Study Area

The study area corresponds to the north-west territorial portion of Peru, where the coastal ENSO occurred during the months of January, February, and March of 2017 (Figure 1). During these months, the strong warming that developed in the Eastern Tropical Pacific Ocean (abnormal warmth) produced unusually intense rain storms in the area, compared to the 1997–1998 and 1982–1983 ENSO events [25].

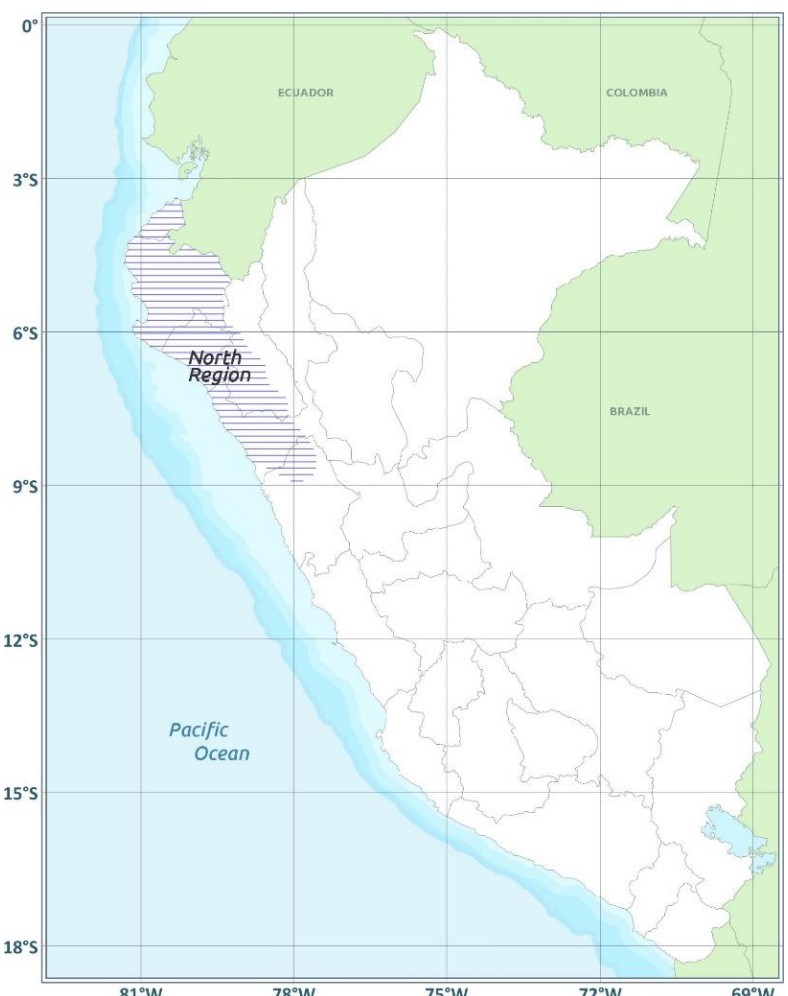

**Figure 1.** Location of the study area.

### 2.2. Methodology

For the development of this study, the oceanic indices that define the occurrence of ENSO have been taken into consideration. Data from different indices correspond to the years 2016 and 2017 (January, February, and March) (Appendix A; Tables A1–A3), in order to determine which of these indices is the most appropriate to define the occurrence of the coastal ENSO. It is likewise with the implications of the ITCZ in the generation of rainfall. In Figure 2 a diagram of the methodological procedure is show.

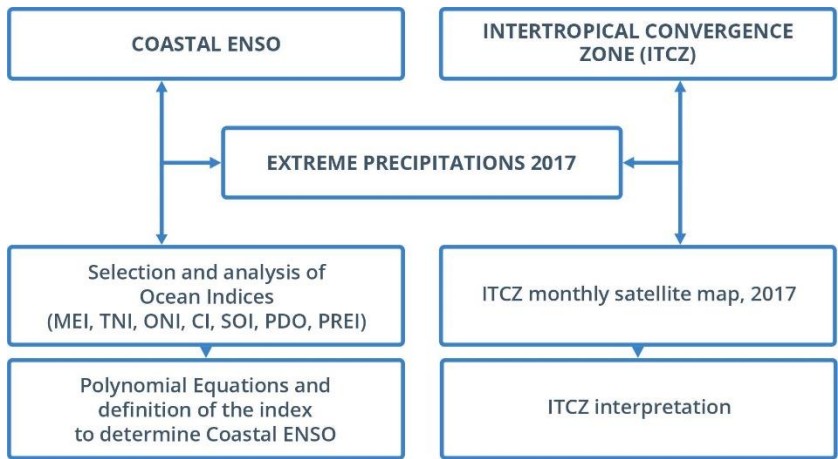

**Figure 2.** Diagram of the methodological process.

2.2.1. ENSO Data and Index Analysis

For the ENSO 2017 analysis, data from the ENSO's regional indices have been used, which is a standardized measure where the Equatorial Pacific is divided into regions, from which the presence of El Niño or La Niña is deduced. To analyze the behavior of the different types of ENSO index, polynomial equations have been used. The polynomial equation can be expressed in Equation (1):

$$f(x) = a_n x^n + a_{n-1} x^{n-1} + \ldots + a_3 x^3 + a_2 x^2 + a_1 x + a_0 \tag{1}$$

where:

- $a_n$, $a_{n-1}$, and $a_0$ are real coefficients (numbers).
- $a_n$ is different than zero.
- The exponent $n$ is a positive integer that represents the degree of the equation.
- $x$ is the unknown variable to be found.

To determine the goodness of fit of each model, the coefficient of determination ($R^2$) was used, which ranges from 0 to 1. The closer its value is to 1, the greater the fit of the model to the variable that is being explained. Conversely, the closer to zero, the less tight (unreliable) the model will be. For the particular case of this investigation, seven indexes were used, corresponding to the years 2016 and 2017 (years in which the ENSO Modoki and the La Niña (LNSO) Modoki were presented: The Modoki ENSO Index, the Trans Niño Index, the Oceanic Niño Index, the Coastal Index (1 + 2), the Southern Oscillation Index, and the Pacific Equatorial Regional Index. In the case of the Pacific Decadal Oscillation (PDO) index, this has been evaluated from 1976 to 2019 (without calculating any equation) to determine its correspondence with the occurrence of Niño Modoki.

Modoki ENSO Index (MEI)

There have been ENSO events that are not part of the evolution of El Niño, which involve ocean-atmosphere coupled processes that during their evolution present a unique pattern of tripolar pressure at sea level, analogous to the Southern Oscillation in the case of El Niño. Therefore, the total event is called "Modoki ENSO", whose presence significantly influences SST and precipitation in many parts of the world [1]. La Niña Modoki events are observed using the negative index of Modoki ENSO [1]. The data used are available at: http://www.jamstec.go.jp/frsgc/research/d1/iod/modoki_home.html.en (access time 21 July 2019). The equation to calculate the MEI Index is shown in Equation (2) [1]:

$$\text{MEI} = TSMA_{BOX\text{-}A} - 0.5 * TSMA_{BOX\text{-}B} - 0.5 * TSMA_{BOX\text{-}C} \tag{2}$$

Trans Niño Index (TNI)

The TNI index is the one that best represents the occurrence of coastal ENSO. The data used are available at: www.esrl.noaa.gov/psd/data/climateindices/List/#TNI (accessed on 21 July 2019). The TNI index measures the gradient of SST anomalies between the Central and Eastern Equatorial Pacific. When the SST gradient is particularly large (for example, positive anomalies in the ENSO 4 Region and negative anomalies in the ENSO 1 + 2 Regions).

Oceanic Niño Index (ONI)

The Oceanic Niño Index (ONI) is the standard that the National Oceanic and Atmospheric Administration (NOAA) uses to identify warm (El Niño) and cold (La Niña) events in the Tropical Pacific Ocean. The ONI Index uses the same region as the Niño 3.4 Index. El Niño 3.4 is negatively correlated with the Southern Oscillation Index (SOI) detailed further down, that is, when the mean SST in the Pacific is high, the SOI is generally low. The data from the ONI Index have been used to determine its correlation with the presence of coastal ENSO. The ONI data are available at: https://www.esrl.noaa.gov/psd/gcos_wgsp/Timeseries/Data/nino34.long.data (accessed on 21 July 2019).

ONI index values are normalized according to a statistical series for a single fixed base period of 30 years. There will be multiple-centered 30-year base periods that will be used to define the ocean index (as a deviation from the average or "anomaly"). The NOAA considers El Niño conditions to be present when the ONI index is +0.5 or higher, indicating warmer than normal. When the ONI index is −0.5 or lower, it indicates that the region is colder than normal and suggests the occurrence of La Niña. The methodology for the reconstruction of the SST is described in [26].

Coastal Index (CI)

The location of this index is in the northwestern part of Peru. The data used are available at: https://www.esrl.noaa.gov/psd/gcos_wgsp/Timeseries/Data/nino12.long.anom.data. (accessed on 21 July 2019). This index corresponds to the averaged area of SST of 0°–10° S and 90°–80° W, covering to the north coast of Peru. Values are calculated as the three-month running mean of the SST anomaly obtained in real time from NOAA Extended Reconstructed SST, minus the 1981–2010 climatology.

Southern Oscillation Index (SOI)

Data from the SOI have been used to determine its correlation with the presence of coastal ENSO. SOI data are available at: https://www.cpc.ncep.noaa.gov/data/indices/soi (accessed on 22 July 2019). The SOI or barometric difference is a standardization based on the differences in sea level pressure observed between Tahiti and Darwin (places in the southeast Pacific such as Easter Island, Darwin in Australia, or Jakarta in Indonesia). Its calculation can be performed using Equation (3).

$$\text{SOI} = 10\frac{P_{dif} - P_{difv}}{SD\left(P_{dif}\right)} \tag{3}$$

where:

$P_{dif}$ = (Tahiti average sea level pressure for a month) − (Darwin average sea level pressure for one month)

$P_{difv}$ = Long-term average of $P_{diff}$ for the month in question

$SD\left(P_{dif}\right)$ = Long-term standard deviation for the month in question

SOI ranges from −35 to +35, and it is calculated based on months and periods of one year. Negative SOI values indicate episodes of the El Niño phenomenon, while positive SOI values indicate episodes of La Niña.

Pacific Decadal Oscillation (PDO) Index

As with the previous cases, data from the PDO index have been used to determine if there is any relationship with the occurrence of coastal ENSO. The data are available at: http://jisao.washington.edu/pdo/img/v1v2PDOComp.png (accessed on 22 July 2019). This oscillation has also been called the Pacific Interdecadal Oscillation (PIO) [27] and the North Pacific Oscillation (NPO) [28], for the explanation of this index in the present study, the term PDO, which is a fluctuation (climatic variability) that resembles ENSO, will be taken with a little more emphasis on the extratropical signal and longer time scales in the Pacific Ocean.

This index does not define the occurrence of ENSO. The generation of this interdecadal oscillation can affect the strength and frequency of El Niño and La Niña. These indices have long periods of occurrence that lasts from 20 to 30 years in the cold or warm phases, much longer than the El Niño Oscillation. ENSO substantially influences the tropical climate, while the PDO affects the North Pacific and the North American continent [27].

The physical mechanisms of PDO are not well understood; reference [29] indicate that PDO modulations are linked to the atmospheric response of tropical SST anomalies, manifested in the local Hadley circulation and the Walker local circulation in low latitudes, and the Rossby wave train in the extratropics, including the Pacific-North American Pattern (PNA) in the northern hemisphere.

Pacific Regional Equatorial Index (PREI)

The oceanic indices developed and currently used do not define the presence of a coastal ENSO. For this reason, the creation of another index called the Pacific Regional Equatorial Index (PREI) has been proposed in this study, using the set of data from the ENSO Regions 4, 3.4, 3 and 1+2, whose standardization will allow to define or interpret Coastal ENSO in an optimal way. The data are available at: https://www.esrl.noaa.gov/psd/gcos_wgsp/Timeseries/Data (accessed on 22 July 2019).

2.2.2. Data and Analysis of the Inter-Tropical Convergence Zone (ITCZ)

The ITCZ was another of the factor that caused the extreme rains in northern Peru in 2017. The ITCZ is a band or convective band of low pressures that is not uniform or continuous located in the equatorial zone that ranges approximately between 5° S and 12° N during July–November, and 5° N–5° S during January to May. It is influenced by the trade winds from the southeast and northeast due to high temperatures; air masses are forced to rise causing abundant cloudiness and heavy rainfall, some accompanied by electrical discharges. Its thickness varies from one site to another, as does its behavior in maritime and continental areas [30,31].

Likewise, satellite images from the Weather Channel (WC) have been used to carry out a qualitative analysis and for comparison to determine the displacement of cloud masses from the ITCZ to the South American continent. The strong convective activity in the ITCZ led to the formation of cloud systems of great vertical development, causing intense rains to the interior of the Ecuadorian and Peruvian coasts [32]. Satellite images were taken for the month of March, the month in which the highest rainfall occurred. In these images, the evolution and displacement of cloud masses were observed, allowing an indirect estimate of rainfall. Images can be obtained from the link: https://weather.com/maps/satellite/southamerica-weather-map (accessed on 22 July 2019).

## 3. Results

### 3.1. Polynomial Equations and Correlation Coefficients of Each Index

Polynomial equations and $R^2$ values from data analysis for each index are shown in Table 1. The coefficient of determination, denoted $R^2$ reflects the goodness of fit of a model to the variable to be explained. Table 2 shows pros and cons of the different oceanic indices treated in the present study. The PREI and ONI indexes were those with best fits, being the SOI index the least.

**Table 1.** Polynomial equations and $R^2$ values for each analyzed index.

| Index | Polynomial Equation | $R^2$ |
|---|---|---|
| MEI | $Y = 7 \times 10^{-7}x^5 - 0.0001x^4 + 0.0053x^3 - 0.0684x^2 + 0.2232x + 0.0856$<br>Y: SST value<br>X: It is the period corresponding to the months | 0.90 |
| TNI | $Y = 1 \times 10^{-5}x^5 - 0.0004x^4 - 0.0045x^3 + 0.173x^2 - 0.933x - 0.7016$ | 0.95 |
| ONI | $Y = 2 \times 10^{-5}x^5 - 0.0015x^4 + 0.0316x^3 - 0.2472x^2 + 0.2189x + 2.4981$ | 0.99 |
| CI | $Y = 2 \times 10^{-5}x^5 - 0.0013x^4 + 0.0234x^3 - 0.1496x^2 + 0.0336x + 1.722$ | 0.89 |
| SOI | $Y = -0.0003x^5 + 0.0187x^4 - 0.3732x^3 + 2.7439x^2 - 3.1877x - 19.397$ | 0.64 |
| PREI | $Y = 0.5175x^2 - 3.3805x + 31.018$<br>X: It is a sequential period representing regions 4, 3.4, 3 and 1 + 2 | 0.99 |

**Table 2.** Comparison of the pros and cons of ocean indices.

| Index | Pros | Cons |
|---|---|---|
| MEI | It is used to determine El Niño Modoki, that is, it only determines SST anomalies in the equatorial Central Pacific. | It does not specifically designed to determine Coastal ENSO |
| TNI | It is used to characterize both the evolution of the El Niño or La Niña event | It does not specifically designed to determine the Coastal ENSO |
| ONI | Is NOAA's primary indicator for monitoring Canonical El Niño and La Niña | Not specifically designed to determine Coastal ENSO |
| CI | It is useful, when a Canonical ENSO is presented | It does not specifically designed to determine the Coastal ENSO |
| SOI | It is useful for determining El Niño and La Niña episodes | It is the least suitable for determining the presence of Coastal ENSO |
| PREI | It allows to define or interpret the Coastal ENSO in an optimal way | It is required to differentiate between the presence of a Canonical ENSO from the presence of a La Niña Modoki "LNSO" |

The value of the coefficient of determination $R^2 = 0.99$ obtained in the case of the PREI index is the result of using the polynomial equation of the order "2" and the value of $R^2 = 0.99$ indicates that the model fits the data, that is, the differences between the observed values and the calculated values are small and not biased. This reflects the quality and goodness of fit of the model. In the next sections, a detailed analysis of each index is provided.

The data used correspond to the years 2016 and 2017, with the exception of the PREI index, which only uses the data for the month of January 2017.

### 3.1.1. Modoki ENSO Index (MEI)

In 2016 (January through March), an increase in the sea surface temperature of 0.30 °C was observed, indicating the occurrence of Modoki ENSO. Later on, sea temperatures decreased to values as low as −0.60 °C in 2017 (January through March), leading LNSO (Figure 3). Surprisingly, this large-scale oscillation from warmer to colder SST and in the same central Equatorial zone of the Pacific Ocean clearly defines the presence of Modoki ENSO and Modoki LNSO, whose process and sequence will also serve to define the presence of coastal ENSO.

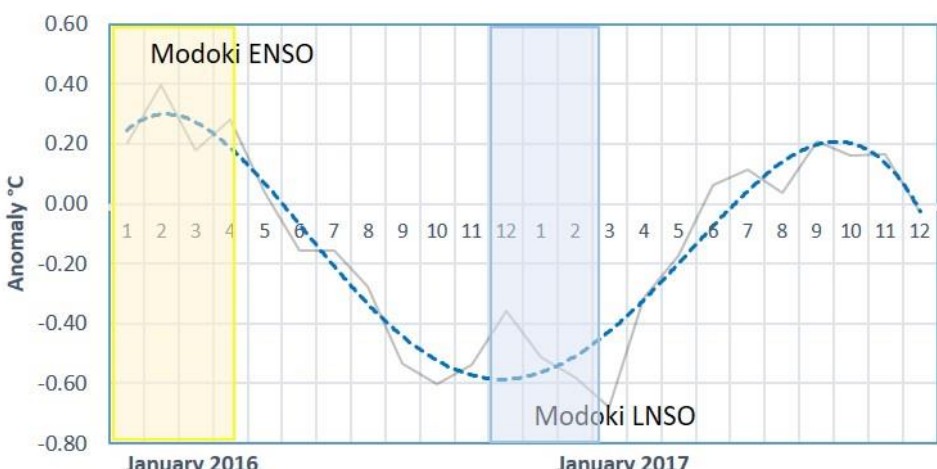

**Figure 3.** Time series of the MEI index, showing the presence of 2016's Modoki ENSO and 2017's Modoki LNSO.

### 3.1.2. Trans ENSO Index (TNI)

The results define the oscillations of the temperature anomalies quite well (Figure 4). There was an increase in sea temperature with a positive anomaly of 1.3 °C in 2017 (January through March). This index regularly represents the occurrence of coastal ENSO.

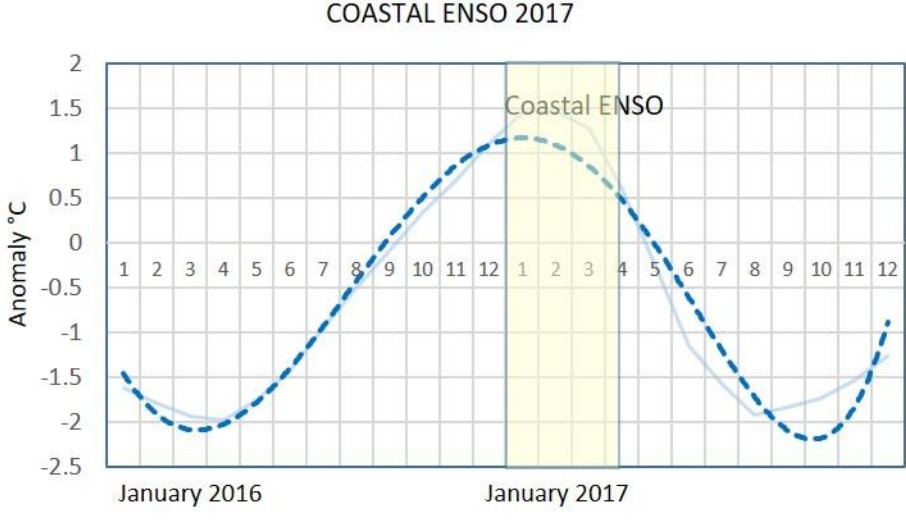

**Figure 4.** Time series of the Trans ENSO Index (TNI), for the years 2016 and 2017. In 2017, a positive phase was observed, indicating the presence of coastal ENSO.

### 3.1.3. Oceanic ENSO Index (ONI)

This index uses data from Region 3.4 (central Pacific zone) to define the ENSO and LNSO, and it is currently being used by NOAA. There was an increase in temperature of 2.5 °C during 2016 (January through March). In this same year (September through august), there was a decrease in temperature with an ONI value of −0.7 °C, indicating the presence of Modoki LNSO (Figure 5). This index adequately represents the presence of Modoki ENSO.

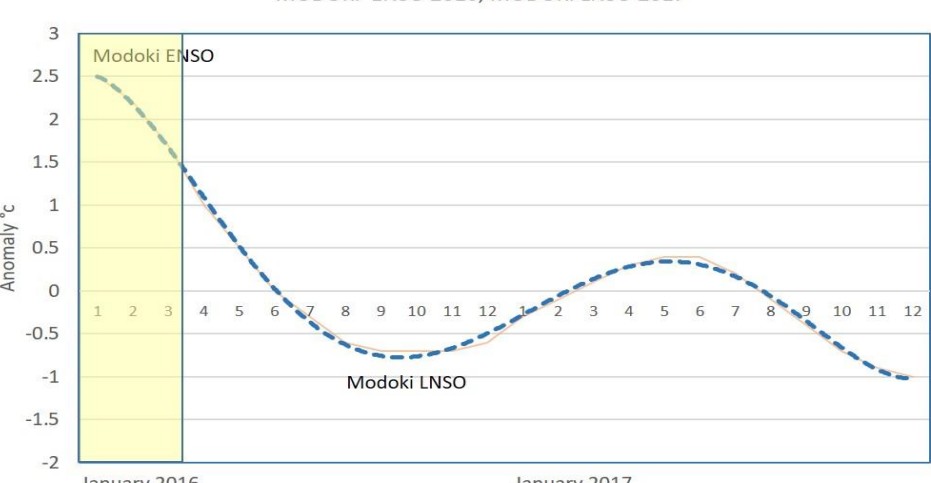

**Figure 5.** Oceanic ENSO Index (ONI) values, for the years 2016 and 2017. In the first months of 2016, high values (2.5 °C) were observed, indicating the presence of Modoki ENSO, although it had no incidence in northern Peru. For the first months of 2017, low values were observed, indicating the presence of Modoki LNSO.

### 3.1.4. Coastal Index (CI) (Region 1+2)

As can be seen from Figure 6, an average increase of SST of 1.5 °C and, therefore, this index does not contribute to establish the presence of coastal ENSO.

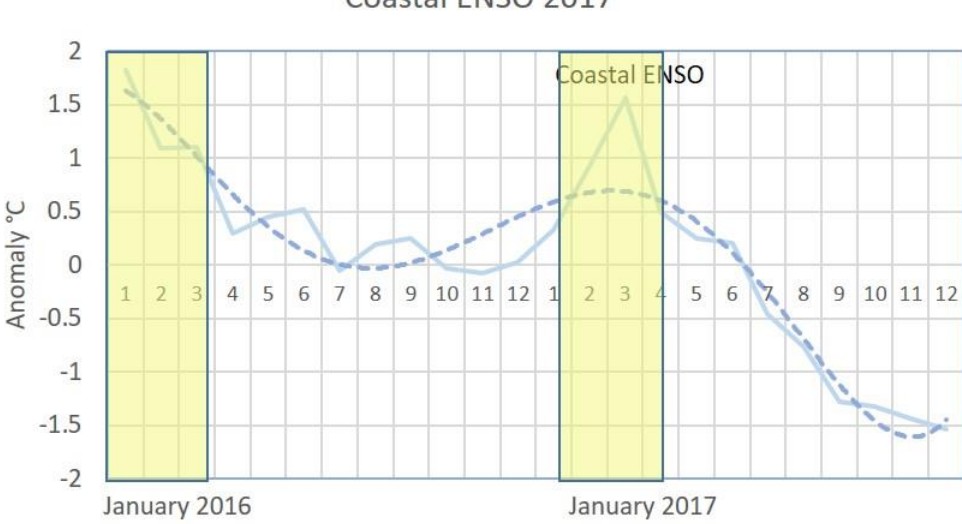

**Figure 6.** Time series of the year 2016 and 2017 using the 1 + 2 index. An increase in SST is observed in both years and in 2017 (March). Therefore, it does not contribute appropriately to establish the present of coastal ENSO.

### 3.1.5. Southern Oscillation Index (SOI)

The SOI is based on atmospheric pressure. Figure 7 illustrates data corresponding to the months of January to March (2016 and 2017). In 2016, negative values were presented, which is consistent with the presence of warm water, defining the presence of ENSO. For the year 2017 (months from January to March), it can be observed that SOI has an average value of "0", i.e., in a neutral position. Therefore, the information provided by SOI is neither useful nor recommended for determining coastal ENSO.

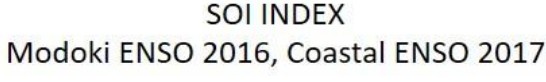
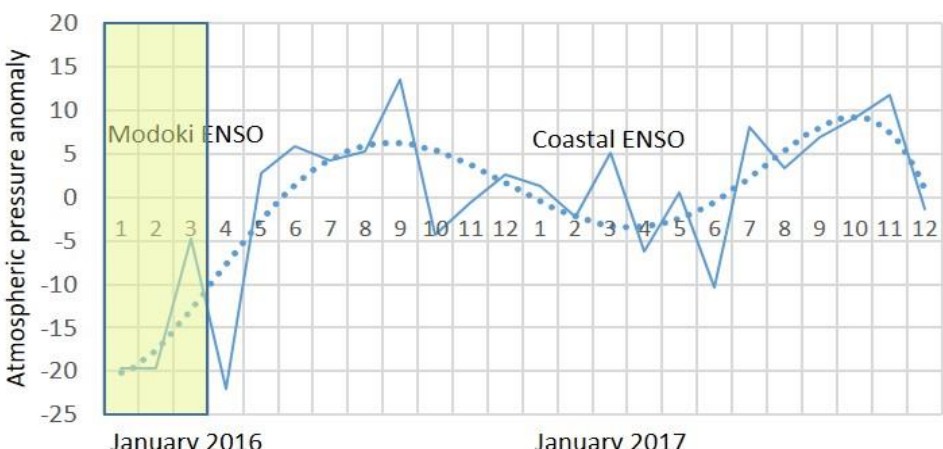

**Figure 7.** Southern Oscillation Index (SOI) showing the presence of 2016's Modoki ENSO, but it is also observed that for the year 2017, it does not adequately define the presence of Modoki LNSO.

### 3.1.6. Pacific Decadal Oscillation (PDO) Index

This index has been used to investigate the influence of the PDO on the occurrence of 2016's Modoki ENSO and 2017's coastal ENSO. Data have been taken from 1976 to 2018, where positive and negative phases can be observed; the longest positive phase was from 1976 to 1988 (12 years), from this date the cycles of both positive and negative phases have had a minor cyclicality of between 4 to 6 years. During the positive phase of 2013–2018, there was a maximum average temperature increase of the order of 1.6 °C (highest peak presented in 2015–2016), coinciding with the generation of ENSO Modoki (Figure 8).

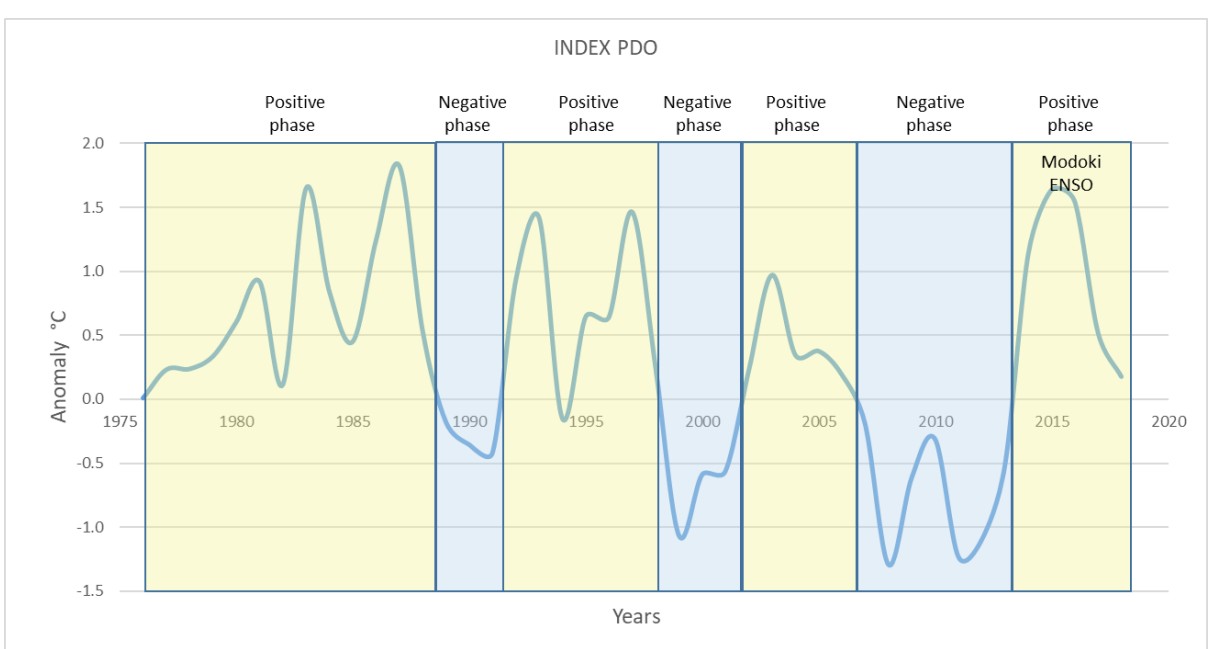

**Figure 8.** Time series of the phase change in the Pacific Decadal Oscillation (PDO), from 1976–2018. The values are annual averages of the anomalies. Positive phases indicate warm years and negative phases indicate cool years.

### 3.1.7. Pacific Regional Equatorial Index (PREI)

Based on the analyzes carried out of the different indices that define the different types of occurrence of ENSO, an index has been defined consisting of the set of data from regions 4, 3.4, 3 and 1+2, which allows us to identify the coastal ENSO for each month and using the corresponding second degree polynomial equation. This index is the one that best represents the evolution of ENSO, in terms of the coastal ENSO. Figure 9 shows a graph of the occurrence of Modoki LNSO and coastal ENSO, during the months of January to March of 2017. In this study we only analyze the behavior of the ENSO Modoki and Niña Modoki (2016–2017) and for the months of January to March, (months where the ENSO phenomenon generally occurs); therefore, we do not intend to generalize for other periods.

a)
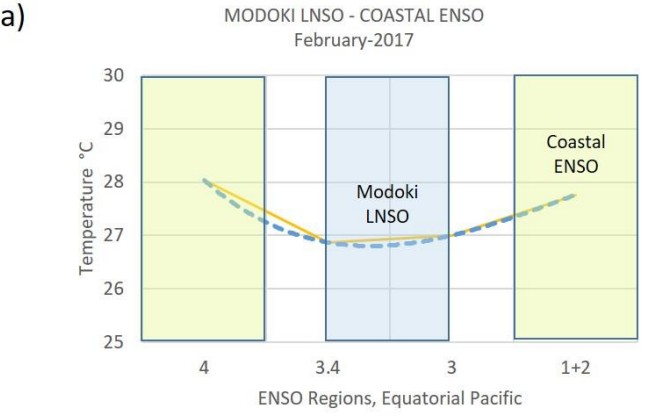

b)
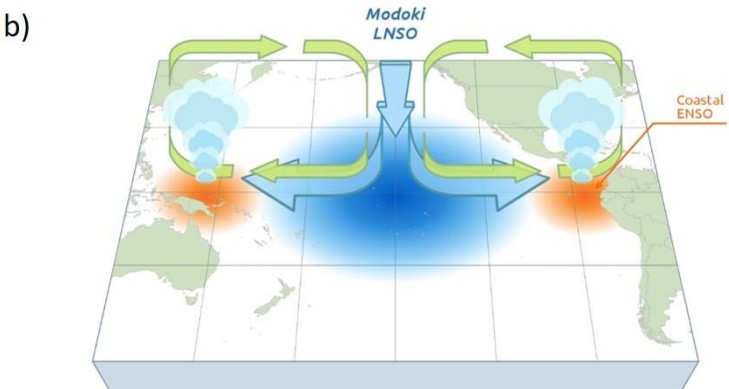

**Figure 9.** (**a**) PREI index, conjunction of regions 4, 3.4, 3, and 1 + 2. The curve shows the presence of Modoki LNSO in the central region (3.4 and 3). On the right margin the eastern region (1 + 2), coastal ENSO is shown. (**b**) Map of the Pacific Equatorial Region, showing the location of the Modoki LNSO and the coastal ENSO.

### 3.1.8. La Niña Modoki (LNSO) and Coastal ENSO Analysis

For the month of January of 2017 (Figure 10a), in region 3.4 and 3, a low value of the SST (25 °C) is observed, indicating the presence of Modoki LNSO and region 1+ 2 also presents a low SST value (25.8 °C). For the month of February 2017 (Figure 10b) in regions 3.4 and 3, a slight increase in SST (26.8 °C) was observed and an increase in SST (27.9 °C) in regions 1 + 2, indicating the presence of coastal ENSO. For the month of March of 2017 (Figure 10c), it can be observed that the temperature had an increase of (28.7 °C) in region 1 + 2, also indicating the presence of coastal ENSO.

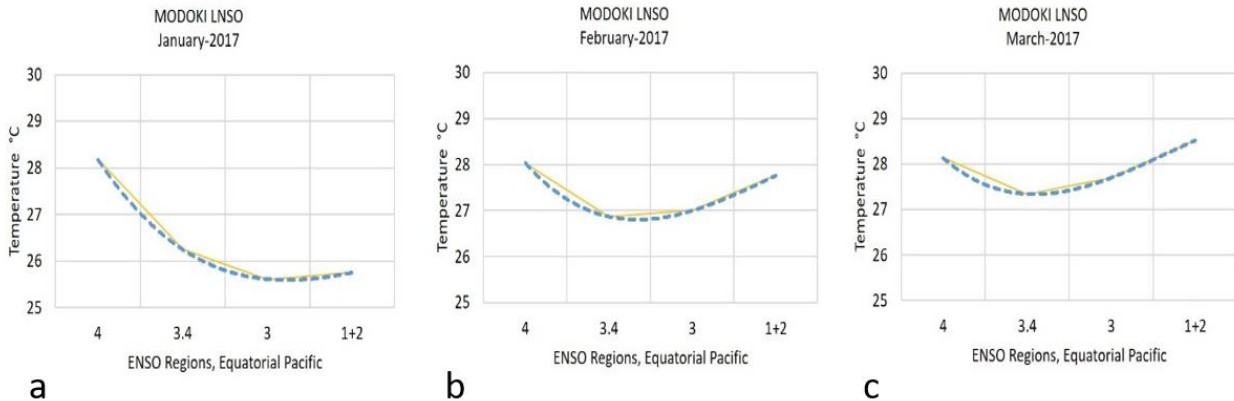

**Figure 10.** Pacific Regional Equatorial Index (PREI). (**a**) January 2017, in regions 3.4 and 3, a low value of the SST (25 °C) is observed, which indicates the presence of the Modoki LNSO and the region 1 + 2, also presents a low value of the SST (25.8 °C). (**b**) February 2017, in regions 3.4 and 3, a slight increase in SST (26.8 C) is observed, but in region 1 + 2 a greater increase in SST (27.9 °C) is observed, indicating the presence coastal ENSO. (**c**) March 2017, in region 1 + 2, an increase in SST (28.7 °C) was observed, in relation to the month of February and presenting the greatest effects of the Coastal ENSO.

In 2017 (Figure 10), if we focus on the values of the region 1 + 2, a sequential SST increase can be observed during the months of January, February, and March (of 25.8, 27.9, and 28.7, respectively) and this elevation of the SST in zone 1 + 2 represented the occurrence of coastal ENSO in the northern zone of Peru. Using together all the indices of the Central Pacific Region, better information is obtained to define the Modoki LNSO and, therefore, determine the occurrence of coastal ENSO, the reason for this study.

### 3.1.9. Demonstration of the Use of the PREI Index

Projection of TSM values (zone 1 + 2) for the months of February and March 2017

To determine the polynomial equation, the SST data (regions 4, 3.2, 3, 1 + 2) corresponding to the month of January have been used, obtaining the following equation:

$$Y = 0.5175x^2 - 3.3805x + 31.018 \tag{4}$$

where:

Y = SST value
X = Variable considered as a period that represents the oceanic regions (4, 3.4, 3 and 1 + 2)

### 3.1.10. Projection for the Month of February

Taking the observed temperature values for the month of January and considering the value of X = 5, the projection for the month of February of zone 1 + 2 is obtained. The value obtained is 27.05 °C, which would indicate the increase in SST, and compared to the observed value of 27.76 °C, there is a relative difference of −0.71 (Table 3) (Figure 11).

**Table 3.** Observed data for the month of January and the projected value for zone 1 + 2 (months February and March).

| Region | Period "X" | "Y" °C Observed (January) | "Y" °C Calculated (January) | Residues °C |
|---|---|---|---|---|
| 4 | 1 | 28.18 | 28.16 | −0.02 |
| 3.4 | 2 | 26.25 | 26.33 | 0.08 |
| 3 | 3 | 25.61 | 25.53 | −0.08 |
| 1+2 | 4 | 25.75 | 25.78 | 0.03 |
|  |  | Observed (February) | Projected (February) |  |
| 1+2 | 5 | 27.76 | 27.05 | −0.71 |
|  |  | Observed (March) | Projected (March) |  |
| 1+2 | 6 | 28.52 | 29.37 | 0.84 |

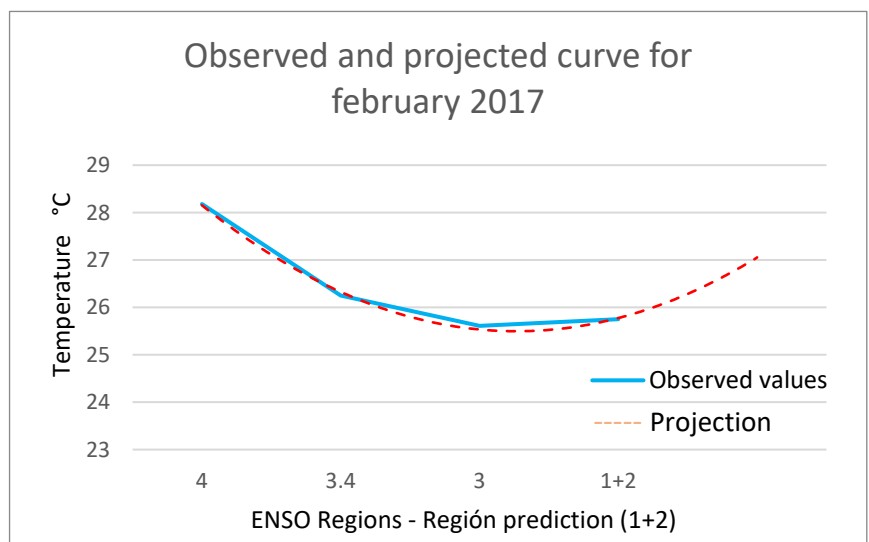

**Figure 11.** Example of the sea surface temperature (SST) projection for zone 1 + 2 (Niño Costero) for the month of February.

### 3.1.11. Projection for the Month of March

Considering the value of X = 6, we have the projection for the month of March. The value obtained is 29.37 °C, the high value of which would indicate a greater increase in SST, which compared to the observed value of 28.52 °C has a difference of 0.84 °C, which would also indicate an adequate approximation and the usefulness of the equation for predictive purposes (Table 3) (Figure 12)

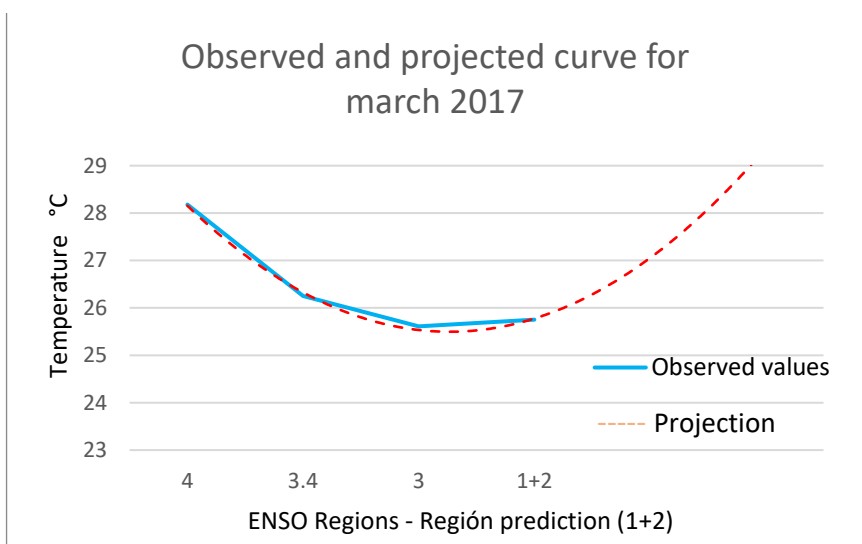

**Figure 12.** Example of the SST projection for zone 1 + 2 (Niño Costero) for the month of March.

### 3.2. Spatial-Temporal Distribution of the ITCZ

In addition to the effects generated by the occurrence of the coastal ENSO, there have also been additional effects from the ITCZ, an area in which convergence between the air masses of the northern and southern hemispheres has been generated. The position of the ITCZ is normally located in the equatorial zone, but it moved further to the south (border area between Brazil and Peru), generating the heavy rainfall that occurred in the country.

A qualitative observation and comparison analysis have been carried out using the images provided by the Weather Channel (WC). In these images, the displacement of cloud masses from ITCZ can be observed. The satellite images for the month of March 2017 (Figure 13a–d) show masses of clouds moving over the north of Brazil and towards the Peruvian territory (central and northern part of Peru). The analysis of these cloud bands shows a connection of a cyclonic circulation with the presence of precipitation, further increasing the precipitation generated by coastal ENSO.

When ENSO occurs, all equatorial convection moves to the east, changing the position of the atmospheric circulation (Walker cell), creating a descending cell over the Atlantic Ocean, and depending on the intensity there may be inhibition in the formation of clouds and consequently the deficiency of rainfall in the ITCZ region [33] In the case of coastal ENSO, there have been thermodynamic patterns or teleconnections of atmospheric circulation (high pressure areas) over the tropical region (equatorial Pacific basin towards the equatorial Atlantic basin), and in the same way the trade winds have contributed to displace the band of clouds from the ITCZ to the northeastern part of Brazil and reaching areas of Peruvian territory.

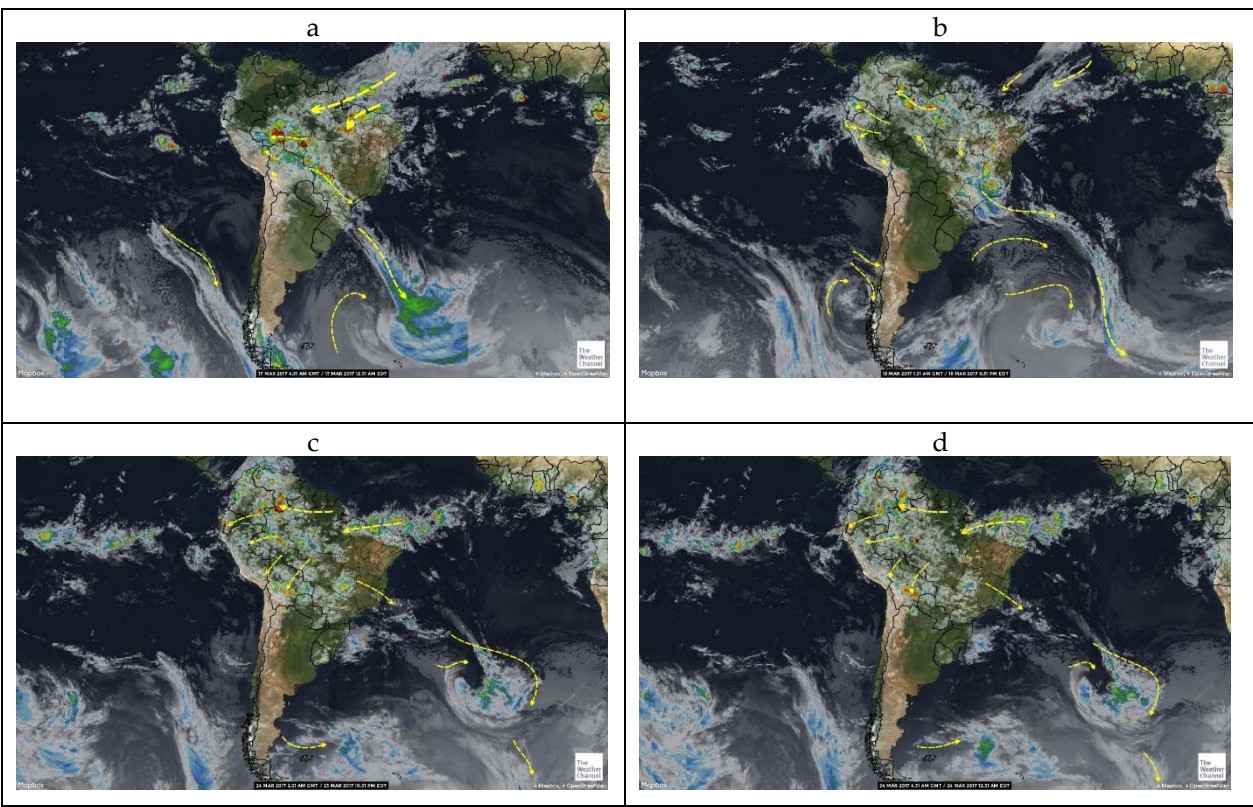

**Figure 13.** Map of the Inter-Tropical Convergence Zone (ITCZ) bands of heavy rainfall and their displacement in the upper part of South America, and their implications for the Peruvian territory, during the month of March; (**a**) March 17, (**b**) March 18, (**c**) March 23, and (**d**) March 24 Reprinted with permission from ref. [34]. Copyright 2017 The Weather Channel.

## 4. Discussion

In the present study, two main sources that generated heavy rainfall in Peru are proposed. First, the generation of a type of ENSO called "Coastal ENSO"; and second, cloud bands generated in the ITCZ. Next, we develop the explanation and discussion about the generation process of the type of ENSO identified in northern Peru.

### 4.1. Types of ENSO

In recent years, other types of ENSO have been investigated and identified, such as Modoki ENSO, Modoki LNSO, and coastal events, whose occurrence is not very common and also brings different consequences. Therefore, the understanding of these different types of ENSO has a greater degree of complexity when trying to predict their occurrence and, as a consequence, government responses may be inadequate due to contradictory forecast reports and errors in decision making.

### 4.2. Definition of Coastal ENSO

A preliminary study carried out by Ramírez [35] mentions that the definition of ENSO was a problem that generated uncertainty and affected the perception of risk and the responses were further complicated by the fact that some regions within Peru were experiencing drought prior to the onset of coastal ENSO. These authors also stated that this "surprise" event developed rapidly and without warning and had catastrophic effects in northern Peru (an ENSO that was not documented since 1925), in addition to contradictory forecast reports (USA and PERU).

Takahashi [36] proposed to classify ENSO into two types: "Global" (e.g., 1982–1983 and 1997–1998) and "Coastal" (e.g., 1925 and 2017). As can be seen, not much literature exist on the occurrence of coastal ENSO in Peru. Similarly, studies carried out by the

Multisectoral Commission in charge of the National Study of the"El Niño" Phenomenon ENFEN [11], Heureux et al. [37], and Ramírez [35] proposed to distinguish coastal ENSO from the warm ENSO phase, operationally defining it as "El Niño Costero" based on the seasonal anomaly (SST) of the ENSO 1 + 2 region, the authors also mention that coastal ENSO requires a more in-depth study in terms of its mechanisms and forecast. In the present investigation, based on the analysis carried out, the authors also coincide with the name of "El Niño Costero", mainly due to its form of generation and occurrence.

### 4.3. 2015–2016 ENSO

The extreme ENSOs that occurred in Peru are only based on the 1982–1983 and 1997–1998 events. The occurrence of the 2017 coastal ENSO, with different characteristics was preceded by the occurrence of the 2015–2016 ENSO (the first extreme El Niño of the 21st century) [23], which generated many environmental disasters and dramatic impacts on a scale that are consistent with what is expected for an extreme ENSO. However, their occurrence showed different characteristics in relation to the last episodes, confusing previous understanding of an extreme ENSO.

### 4.4. Oceanic Indexes

There are different types of ocean index that have the purpose of observing the occurrence of ENSO, which have not always been effective when predicting the generation of an ENSO event. Each agency examines a wide range of oceanic and atmospheric anomalies to report their updates internationally. The 3.4 ENSO Region, located in the central Equatorial zone of the Pacific Ocean, is perhaps the most common measurement of ENSO, because this region is strongly coupled with the overlying atmosphere and global teleconnections [24]. However, this ENSO 3.4 index did not allow researchers to foresee the occurrence of the 2017 event.

### 4.5. Analysis of the Different Ocean Indices and Creation of the PREI Index

An analysis of the different existing oceanic indices has been carried out in the present investigation, such as Modoki ENSO Index (MEI), Trans ENSO Index (TNI), Oceanic ENSO Index (ONI), Coastal 1 + 2 Index (CI), Southern Oscillation Index (SOI), and Pacific Decadal Index (PDO), in order to determine which of these indices are the most suitable for monitoring and determining the occurrence of an extreme coastal ENSO. From the results of these analyses, it has been determined that these indices do not have an effective application to specify the occurrence of a coastal ENSO such as that which occurred in 2017. For this reason, a new index called the Pacific Regional Equatorial Index (PREI) has been developed in this investigation to specifically determine a coastal ENSO.

### 4.6. Causes of Coastal ENSO Generation

In relation to the causes that generate a coastal ENSO, few studies have been carried out such as that by Takahashi [36], who proposed that the occurrence of coastal ENSO is due to the north winds, which has a crucial role for its development. Likewise, a study carried out by Peng [38] refers that the descending Kelvin waves caused by strong westerly winds over the Equatorial Pacific, together with the anomalous coastal winds from the north, contribute to the formation of the extreme coastal ENSO. Similarly, Peng [38] concluded that the combined effect of local winds and Equatorial Kelvin waves caused the 2017 extreme coastal ENSO. Likewise, Garreaud [18] mentions that the cause was a sustained and forced weakening of the western tropospheric flow that impinges on the subtropical Andes and leads to a relaxation of the southeastern winds off the coast, which in turn may have warmed the eastern Pacific, along the weakening of the outcrops in a near-shore band and decreased evaporative cooling farther from shore.

In this study, the authors differ with the previous explanations, since according to the analysis and results obtained, it has been determined that the occurrence of a coastal ENSO is due to a process of succession of events (Modoki ENSO and Modoki LNSO). In such a

way we postulate that the Pacific Decade Oscillation (PDO) in its positive phase (Figure 8), has given rise to coupled ocean-atmosphere processes generating ENSO Modoki and La Niña MODOKI, and the coastal ENSO is a sub-process of La Niña Modoki. the 2017's Modoki LNSO-lower part (Figure 9).

Takahashi [36] studied the coastal event of 1925, suggesting a possibility that the concurrent LNSO conditions in central Pacific could be in favor of atmospheric convection off Peru, reducing atmospheric stability. In this study, the authors partially agree with this hypothesis, changing the term LNSO to Modoki LNSO, since this type of event was the one that generated the coastal ENSO. Given these assertions, we go on to explain some of the causes that generated the Modoki events.

### 4.7. Causes of the Generation of Modoki ENSO/Modoki LNSO

In a study carried out by Ashok [1], the authors state that the Modoki El Niño involves coupled ocean-atmosphere processes that include a unique tripolar sea pressure level pattern, analogous to the Southern Oscillation in the case of El Niño and state that the weakening of the equatorial east winds are related to the zonal weakening of the sea surface temperature gradient that leads to a greater flattening of the thermocline, and that this could be a cause of the more frequent and persistent occurrence of the ENSO Modoki event.

Similarly, Behera and Yamagata [39] mention that Modoki El Niño/Modoki La Niña have a coupled ocean-atmosphere variability in the tropical Pacific Ocean and that the SST anomalies are coupled with a double cell in the Walker anomalous circulation, with upward movement in the central parts and downward movement on both sides of the basin of the tropical Pacific Ocean. Furthermore, they consider that Modoki events are not seasonally locked into a single seasonal cycle like El Niño/La Niña events.

Likewise, Lestari et al. [40] concluded that after an extreme phenomenon of El Niño 2015–2016, a Modoki La Niña event occurred, which had a short duration and was associated with a pattern of SST anomalies in the Indo-Pacific region, with a strengthening of easterly winds in the central tropical Pacific Ocean. The cooling (warming) of SST activated the zone of divergence (convergence) and the downward (upward) atmospheric motion and suppressed (enhanced) convective process in the central tropical Pacific Ocean region.

In the present study on the generation of Modoki El Niño, we agree with the aforementioned authors regarding the explanation of the process, which is due to the ocean-atmosphere coupled process that included a unique tripolar sea pressure level pattern, as well as the greater flattening of the thermocline and that do not present a seasonal cycle like canonical El Niño events.

As in the El Niño Modoki generation, the Niña Modoki involves processes of the Atmospheric Ocean in the tropical Pacific Ocean coupled to two cells of the Walker circulation and defining a tripolar pattern; that is, two upward movements (one at the end of region 4, and another in region 1+2) and one downward movement in the middle (regions 3.4 and 3) (Figure 9). This process and the temperature differences of the SST, can be observed in February 2017 (taken as a median); that is, at the extremes of the ENSO Regions (4 and 1+2) there are high SST values (28.03 °C and 27.8 °C) and in the middle part of the ENSO Regions (3.4 and 3) there are low values (26.8 °C) (Figure 9).

### 4.8. Decadal Cause

Other studies carried out on the generation of Modoki El Niño explain that they are generated by PDO, such as that carried out by Weng [41], who state that SST gradients result in a two-cell Walker circulation anomalous over the tropical Pacific, with a humid region in the central Pacific and that Modoki El Niño has a large decennial bottom (PDO), while canonical El Niño is dominated by interannual variability.

Li [42] mentions that, in addition to the atmosphere-ocean coupling on intra-seasonal and inter-annual time scales, inter-decennial PDO variations and longer time scales can play an important role (and sometimes a crucial role) in determining the intensity of El Niño. Likewise, Dong [29] indicate that PDO modulations are linked to atmospheric responses

and tropical SST anomalies, manifested in the local Hadley and Walker circulations in low latitudes, and the Rossby Wave Train in the extra-tropics, including the Pacific North American pattern (PNA) in the northern hemisphere.

Likewise, Diaz et al. [43] argue that, in regional climates around the world, there have been important changes on a 10-year scale in the general pattern of SST and in the general atmospheric circulation, pointing to the complex interaction between the Canonical ENSO system. Similarly, Magee [44] states that PDO can modulate the genesis of tropical cyclones (TC) during ENSO and Modoki ENSO events, in favor of northeast/southwest modulations typical of positive/negative PDO events, and that Modoki ENSO provides an important piece of the puzzle in understanding where and when the genesis of TC can occur.

In this study, we agree with the assertions mentioned by the previous authors in relation to the fact that the PDO generates the Modoki ENSO. Nevertheless, according to our analysis (from the years 1976–2018), where positive and negative phases can be observed, the longest positive phase was from 1976 to 1988 (12 years), from this date the cycles of both positive and negative phases have had a minor cyclicality of between 4 to 6 years. A positive phase began in 2013, reaching its highest peak (annual anomaly of 1.5 °C) in the years 2015–2016, dates during which Modoki El Niño was generated or appeared (Figure 8).

Due to the action of the PDO, the surface winds change, generating anomalous wind patterns to propel the ocean in the mid-latitude of the Pacific Ocean. The coastal ENSO is a sub-process of La Niña Modoki, that is, it is a consequence of another larger event (PDO) and also deserves a more in-depth investigation which is beyond the scope of this study. In Figure 8, the behavior of the oscillation of the PDO Index can be observed as explained.

The El Niño Southern Oscillation (ENSO) and the Pacific Decadal Oscillation (PDO) are large-scale climatic phenomena that affect atmospheric and oceanic teleconnections and, therefore, trigger meteorological phenomena at different temporal and spatial scales, both in tropical and extratropical regions. During 11 years (2007–2017) the influence of ENSO events (Canonical and Modoki- in their El Niño, La Niña and Neutral modes) and PDO phases (cold and warm) on meteorological conditions was analyzed [45].

ENSO teleconnections may depend on the phases of the PDO and the characteristics of warming over the central and eastern tropical Pacific, furthermore, the strength of ENSO teleconnections appears to depend on patterns of SST anomalies over the pacific tropical [46].

The frequency and strength of ENSO Modoki events have increased considerably since previous decades (1970–2010), resulting in the need to review climate impacts of varying magnitude from ENSO Modoki, and like ENSO Canonical, ENSO Modoki it also induces a considerable impact in the North Pacific (Atlantic) region [47].

### 4.9. Analysis in the Generation of an Anticyclone

Rodríguez et al. [19] proposed that the probable cause of the anomalous storms generated by the 2017 ENSO is related to the combination of an especially intense wet season over the Central Andes, related to a deep and long-lasting anticyclone located adjacent to the Chilean coast, and the unusual development of warm water off the coast of Peru from El Niño Region 1 + 2. What was explained in the previous study is partially compatible with the occurrence of the ITCZ proposed in this investigation. According to the analysis carried out, we propose that the presence of coastal ENSO in the ITCZ would have generated areas of low pressure, favoring the formation and rise of large masses of humid air, turning into extensive clouds that produced the heavy rainfall in the central and northern areas of Peru. The position of the ITCZ had a variability and moved more towards the southwest of South America (January to March).

## 4.10. Climate Variability-ENSO

The incidence of climate variability in the generation of ENSO is not yet clear, given the existing complexity of the planet's ocean-atmosphere interaction, and only some theories are mentioned in this regard. Tsonis [48], for example, mentions that climate change and ENSO converge in the conclusion that changes in ENSO statistics occur in response to fluctuations in climate (temperature), and the discovery of a correlation between the frequency of El Niño recurrence and temperature change. Similarly, another study carried out by Chowdhury [33], in which they used regional climate models to evaluate future variability of ENSO, indicate that ENSO is correlated with both global and regional climate change scenarios, and that the temperature will continue to increase. In addition to the changing climate, the interannual variability of ENSO would further compound the problems.

## 4.11. Limitations of Data and Methods

In the present study we have carried out an evaluation of the oceanic indices to determine the most appropriate index and to be able to determine a coastal ENSO We are only evaluating two years (2016–2017), because they are the years in which ENSO Modoki, Niña Modoki and "2017 Coastal ENSO" and according to available bibliographic information on coastal ENSO in Peru, there are only two events, a coastal event that occurred in 1925 and the last coastal event in 2017, so there is not much information to make a future evolution or prediction of a coastal ENSO. However, in the case of the PDO index we are evaluating from 1976 to 2018 because we postulate that it could be the cause in the generation of the coastal ENSO.

## 5. Conclusions

According to the results obtained in this study, the heavy rainfall that occurred in the northern part of Peru has been caused by the presence of a special type of ENSO, called "Coastal El Niño" (*El Niño Costero*), as well as the formation of extensive masses of humid air generated in the ITCZ and, therefore, Peru would be affected by three types of meteorological phenomena (in relation to rainfall extremes): the "Canonical El Niño", the "Coastal El Niño", and the ITCZ. It has also been observed that the occurrence of the "Coastal El Niño" is directly related to the occurrence of El Niño Modoki and this, in turn, with the positive phase of the PDO.

To determine the generation and occurrence of "Coastal El Niño", different ocean indices have been analyzed, of which the index that best adapts is the index developed by the authors, called "PREI", through which the probable increases in temperature can be monitored of zone 1+2 that may occur during the months of January to March of a given year and allowing proper management of disaster related to the occurrence of a coastal ENSO. Future research is proposed related to the systematization of the best types of indices that can predict the occurrence of an ENSO, as well as atmospheric modeling to predict and differentiate types of ENSO (Canonical and Modoki) and their variants. Likewise, research is suggested on the behavior of PDO and its probable influence on the generation of "El Niño Modoki".

**Author Contributions:** E.G. designed this research, collected observed data, wrote the first draft and edited the paper; E.I. provided professional guidance, writing- reviewing. All authors have read and agreed to the published version of the manuscript.

**Funding:** This research received partial external funding for the first author from Universidad Nacional San Agustin.

**Acknowledgments:** The authors express special appreciation to Pablo Garcia-Chevesich, for reviewing the article, as well as Steve Gonzales V. and Kenny Gonzales V. for the design of the figures.

**Conflicts of Interest:** The authors declare no conflict of interest.

## Appendix A

**Table A1.** Indices Values, from 2016 to 2017.

| Year | Month | Ocean Indices | | | |
|---|---|---|---|---|---|
| | | **TNI °C** | **ONI °C** | **CI** | **SOI** |
| 2016 | Jan | −1.63 | 2.5 | 1.82 | −19.7 |
| | Feb | −1.80 | 2.2 | 1.09 | −19.7 |
| | Mar | −1.95 | 1.7 | 1.10 | −04.7 |
| | Apr | −1.98 | 1.0 | 0.29 | −22.0 |
| | May | −1.76 | 0.5 | 0.45 | 02.8 |
| | Jun | −1.44 | 0.0 | 0.52 | 05.8 |
| | Jul | −0.97 | −0.3 | −0.06 | 04.2 |
| | Aug | −0.51 | −0.6 | 0.19 | 05.3 |
| | Sep | −0.10 | −0.7 | 0.25 | 13.5 |
| | Oct | 0.32 | −0.7 | −0.03 | −04.3 |
| | Nov | 0.69 | −0.7 | −0.08 | −00.7 |
| | Dec | 1.10 | −0.6 | 0.02 | 02.6 |
| 2017 | Jan | 1.45 | −0.3 | 0.33 | 01.3 |
| | Feb | 1.47 | −0.1 | 0.92 | −02.2 |
| | Mar | 1.23 | 0.1 | 1.56 | 05.1 |
| | Apr | 0.59 | 0.3 | 0.49 | −06.3 |
| | May | −0.27 | 0.4 | 0.25 | 00.5 |
| | Jun | −1.15 | 0.4 | 0.20 | −10.4 |
| | Jul | −1.58 | 0.2 | −0.46 | 08.1 |
| | Aug | −1.93 | −0.1 | −0.76 | 03.3 |
| | Sep | −1.85 | −0.4 | −1.28 | 06.9 |
| | Oct | −1.74 | −0.7 | −1.32 | 09.1 |
| | Nov | −1.55 | −0.9 | −1.44 | 11.8 |
| | Dec | −1.26 | −1.0 | −1.54 | −01.4 |

Source: NOAA, Climate Prediction Center (CPC); www.esrl.noaa.gov/psd/data/climateindices/List/#TNI. https://www.esrl.noaa.gov/psd/gcos_wgsp/Timeseries/Data/nino34.long.data; https://www.esrl.noaa.gov/psd/gcos_wgsp/Timeseries/Data/nino12.long.anom.data; https://www.cpc.ncep.noaa.gov/data/indices/soi (accessed on 21 July 2019).

**Table A2.** MEI Index data, during the years 2016 and 2017.

| AÑO | MES | BOX_A | BOX_B | BOX_C | MEI |
|---|---|---|---|---|---|
| 2016 | 1 | 1.35 | 1.94 | 0.35 | 0.20 |
| | 2 | 1.21 | 1.29 | 0.34 | 0.40 |
| | 3 | 1.11 | 1.33 | 0.53 | 0.18 |
| | 4 | 0.94 | 0.77 | 0.55 | 0.28 |
| | 5 | 0.74 | 0.71 | 0.70 | 0.03 |
| | 6 | 0.65 | 0.74 | 0.87 | −0.16 |
| | 7 | 0.50 | 0.41 | 0.90 | −0.16 |
| | 8 | 0.31 | 0.27 | 0.90 | −0.27 |
| | 9 | 0.22 | 0.56 | 0.96 | −0.54 |
| | 10 | 0.02 | 0.30 | 0.93 | −0.60 |
| | 11 | 0.05 | 0.37 | 0.81 | −0.54 |
| | 12 | 0.14 | 0.29 | 0.71 | −0.36 |
| 2017 | 1 | 0.18 | 0.64 | 0.76 | −0.51 |
| | 2 | 0.23 | 1.08 | 0.53 | −0.58 |
| | 3 | 0.18 | 1.12 | 0.60 | −0.68 |
| | 4 | 0.36 | 0.81 | 0.53 | −0.32 |
| | 5 | 0.43 | 0.58 | 0.63 | −0.17 |
| | 6 | 0.51 | 0.26 | 0.64 | 0.06 |
| | 7 | 0.52 | 0.09 | 0.72 | 0.11 |
| | 8 | 0.43 | −0.12 | 0.90 | 0.04 |
| | 9 | 0.23 | −0.69 | 0.73 | 0.21 |
| | 10 | 0.20 | −0.63 | 0.71 | 0.16 |
| | 11 | 0.08 | −0.81 | 0.63 | 0.17 |
| | 12 | −0.08 | −0.86 | 0.76 | −0.03 |

Source: Japan agency for marine-earth science and technology; http://www.jamstec.go.jp/frsgc/research/d1/iod/modoki_home.html.en (accessed on 21 July 2019).

**Table A3.** PDO Index data, during the years 2000 to 2017.

| AÑO | ENE | FEB | MAR | ABR | MAY | JUN | JUL | AGO | SET | OCT | NOV | DIC | PROM |
|---|---|---|---|---|---|---|---|---|---|---|---|---|---|
| 2000 | −2.0 | −0.8 | 0.29 | 0.35 | −0.1 | −0.4 | −0.7 | −1.2 | −1.24 | −1.3 | −0.53 | 0.52 | −0.59 |
| 2001 | 0.6 | 0.29 | 0.45 | −0.3 | −0.3 | −0.5 | −1.3 | −0.8 | −1.37 | −1.4 | −1.26 | −0.9 | −0.56 |
| 2002 | 0.27 | −0.6 | −0.4 | −0.3 | −0.6 | −0.4 | −0.3 | 0.6 | 0.43 | 0.42 | 1.51 | 2.1 | 0.22 |
| 2003 | 2.09 | 1.75 | 1.51 | 1.18 | 0.89 | 0.68 | 0.96 | 0.88 | 0.01 | 0.83 | 0.52 | 0.33 | 0.97 |
| 2004 | 0.43 | 0.48 | 0.61 | 0.57 | 0.88 | 0.04 | 0.44 | 0.85 | 0.75 | −0.1 | −0.63 | −0.2 | 0.35 |
| 2005 | 0.44 | 0.81 | 1.36 | 1.03 | 1.86 | 1.17 | 0.66 | 0.25 | −0.46 | −1.3 | −1.5 | 0.2 | 0.38 |
| 2006 | 1.03 | 0.66 | 0.05 | 0.4 | 0.48 | 1.04 | 0.35 | -0.7 | −0.94 | −0.1 | −0.22 | 0.14 | 0.19 |
| 2007 | 0.01 | 0.04 | −0.4 | 0.16 | −0.1 | 0.09 | 0.78 | 0.5 | −0.36 | −1.5 | −1.08 | −0.6 | −0.20 |
| 2008 | −1 | −0.8 | −0.7 | −1.5 | −1.4 | −1.3 | −1.7 | −1.7 | −1.55 | −1.8 | −1.25 | −0.9 | −1.29 |
| 2009 | −1.4 | −1.6 | −1.6 | −1.7 | −0.9 | −0.3 | −0.5 | 0.09 | 0.52 | 0.27 | −0.4 | 0.08 | −0.61 |
| 2010 | 0.83 | 0.82 | 0.44 | 0.78 | 0.62 | −0.2 | −1.1 | −1.3 | −1.61 | −1.1 | −0.82 | −1.2 | −0.31 |
| 2011 | −0.9 | −0.8 | −0.7 | −0.4 | −0.4 | −0.7 | −1.9 | −1.7 | −1.79 | −1.3 | −2.33 | −1.8 | −1.23 |
| 2012 | −1.4 | −0.9 | −1.1 | −0.3 | −1.3 | −0.9 | −1.5 | −1.9 | −2.21 | −0.8 | −0.59 | −0.5 | −1.10 |
| 2013 | −0.1 | −0.4 | −0.6 | −0.2 | 0.08 | −0.8 | −1.3 | −1 | −0.48 | −0.9 | −0.11 | −0.4 | −0.52 |
| 2014 | 0.3 | 0.38 | 0.97 | 1.13 | 1.8 | 0.82 | 0.7 | 0.67 | 1.08 | 1.49 | 1.72 | 2.51 | 1.13 |
| 2015 | 2.45 | 2.3 | 2 | 1.44 | 1.2 | 1.54 | 1.84 | 1.56 | 1.94 | 1.47 | 0.86 | 1.01 | 1.63 |
| 2016 | 1.53 | 1.75 | 2.4 | 2.62 | 2.35 | 2.03 | 1.25 | 0.52 | 0.45 | 0.56 | 1.88 | 1.17 | 1.54 |
| 2017 | 0.77 | 0.7 | 0.74 | 1.12 | 0.88 | 0.79 | 0.1 | 0.09 | 0.32 | 0.05 | 0.15 | 0.5 | 0.52 |

Source: http://jisao.washington.edu/pdo/img/v1v2PDOComp.png (accessed on 21 July 2019).

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
