# Peer review of "Determination of a New Coastal ENSO Oceanic Index for Northern Peru"

_climate, doi:10.3390/cli9050071_

Round 1

Reviewer 1 Report

This paper presents to us a developement of a  new index to predict the magnitude of El Niño coastal phenomenon in northern Peru.  Authors proposed a new oceanic index called Pacific Equatorial Regional Index and presented the precision value. The paper reads well. I would like to recommend publication after minor revisions. The paper needs a througly  checking for typo and others (i.e ENSO). Authors mentioned  Pacific Equatorial Regional Index as IREP in abstract but later they used PERI. This should be consistent.  Authors didn't justfiy how the precison value is 1 (it is perfect)? As they compared with other indexes so it is required to mention the pros and cons. 

Reviewer 2 Report

Dear authors,

 The follow comments and suggestions will be consider in order to clarify and  improve your manuscript. 

Line 10-12: The authors mention the main objetive of this study was to determinate and differentiate the occurrence of the traditional ENSO, specifically, "El Niño costero", Could be you change "traditional " per canonical ENSO.

Line 13-14: The authors use the polynomial equation method with data from the different types of "existing Ocean Indices" to determine the occurrence of the ENSO, in that sense, Could you explain "the new" index building with the existing indices?. The Equatorial Pacific indices are remain, the methodology for analysis the 2017 Coastal El Niño by polynomial equations is the differences and contribution for this manuscript, you must be consider change the title: "Determination the occurrence of 2017 El Niño Costero or 2017 Coastal ENSO".

 Line 19-20: The set of the oceanic index proposed characterized the occurrence of The El Niño Costero, the IREP (PERI or PREI) is the analysis of the oceanic indices. 

Line 91: In your manuscript mention that this study seems to understand the Coastal ENSO, but in the line 96 the objetive of this study is to determine the occurrence of ENSO, ... you must be use "2017 Coastal ENSO" in both cases, 

Line 244,247, 295: There area a differences between PERI and PREI in line 377?  There are the "IREP index " in line 539 and line 547, PERI=PREI=IREP?

Line 305: The Modoki LNSO? is very confusing acronym, It is the Modoki ENSO during JFM.

Line 378-380: I strongly suggest do not use a "new" index The PREI, because, the set of indices 4,3.4,3 and 1+2 will use in order to represent the "core" of the ENSO in the Pacific Equatorial Region Index.

Author Response

We attach the file with the responses to the observations.

Reviewer 3 Report

This manuscript tends to show a new index to predict the El Nino Coastal. But I have some question for this manuscript. 1. The main structure is not arranged well. In the Section 2, too many words are put on the general knowledge. All the indexes are all already known for the researcher in the same area. For each of those indexes, one or two sentences is enough. The Figure 2 & 3 & 4 are not necessary to show. 2. In the Table 1, what is x? 3. From Figure 5 to Figure 10, I want to see all the time series (say, from 1980-2020). In here, only two years data are used to fitted the equation, I think it is nonsense. Such fitting within short period (only 2 years) is only a fitting, it has no relation with predicting. In the title, a new index to predict…. In fact, I cannot find any such information in your manuscript. 4. How to define the PRE (Line 377)/IREP? It is not clearly defined. 5. All the results about Figure 13 are also nonsense. It is hard for me to find any useful results.

Author Response

(The authors gave the same response as above.)

Reviewer 4 Report

The review of the manuscript climate-1138853

Important comments

My concern is that an ENSO type called "Coastal ENSO" is being introduced for a single extreme precipitation event in Northern Peru. Please add more information and analysis on this event. I propose to introduce the event in the context of the spatial/temporal anomaly. The anomaly related to the ITCZ needs to be introduced (see below). Please explain better the connection of the event with the decadal cause, this can be (or is) very speculative.

Another concern is related to papers [36] and [37]. The results from the papers need to be presented and widely discussed in the introduction section. Please explain better how your results relate to [36] and [37]. Is there any new literature? The conclusions in L564-566, L570-573, L599-603 and especially L630-636 need to be clearly supported with results.

The authors analyse the behaviour of different types of ENSO indices using Polynomial Equations. The problem with polynomial equations lies in the Fundamental Theorem of Algebra: a polynomial of degree n has exactly n complex roots. The authors fit the polynomials to the data in the period 2016-2017, but outside this period the equations are useless. The authors can remove Table 1 and L119-137. The new problem will appear because the methods section is missing.

Section 3.3 L468-473 Not supported with references. Please change Figure 4 which will show anomalies for each month. The anomalies will give better information about the extreme precipitation in 2017 and the ITCZ. In the paragraph, the authors are essentially discussing a meteorological event on the synoptic time scale and linking it to another meteorological event on a much longer time scale. I suggest rewriting the paragraph, expanding it, and adding more information to support the main idea.

Author Response

(The authors gave the same response as above.)

Round 2

Reviewer 2 Report

Dear authors,

The manuscript is accepted in the present version.

Author Response

The reviewer agrees to the first review

Reviewer 3 Report

The authors modified some points in this version. However, I still have some questions   1. The methods in this manuscript have no relationship with prediction. So all the related words should be deleted from the manuscript. Line 336,337: prediction Line 395: predictor Line 701: forecast  Line 447,543,721: predict   2. In table 1, what is x? Say, the x for MEI is same as the x for PDO? These meaning for all the x should be clarified clearly. 

Author Response

Answer is attached

Reviewer 4 Report

No comments

Author Response

The reviewer agrees to the first review